# Study on the Optimization, Extraction Kinetics and Thermodynamics of the Ultrasound-Assisted Enzymatic Extraction of *Tremella fuciformis* Polysaccharides

**DOI:** 10.3390/foods13091408

**Published:** 2024-05-03

**Authors:** Furong Hou, Shasha Song, Shuhui Yang, Yansheng Wang, Fengjuan Jia, Wenliang Wang

**Affiliations:** 1Institute of Agro-Food Sciences and Technology, Shandong Academy of Agricultural Sciences, Jinan 250100, China; houfurong2010@163.com (F.H.); songshasha125@163.com (S.S.); jessicayang1998@163.com (S.Y.); sdnky_wys@163.com (Y.W.); jfj.5566@163.com (F.J.); 2College of Food Science and Engineering, Shandong Agricultural University, Taian 271018, China

**Keywords:** *Tremella fuciformis* polysaccharides, extraction, ultrasound, kinetics, thermodynamics

## Abstract

In this study, *Tremella fuciformis* polysaccharides (TFPs) were extracted by ultrasound-assisted enzymatic extraction (UAE) at different extraction parameters in order to explore the potential of ultrasound in intensifying the extraction yield. The effects of experimental conditions on the extraction yields were optimized using response surface methodology, with the optimal ultrasonic power of 700 W, temperature of 45 °C and time of 50 min. The kinetic analysis revealed that UAE significantly promoted the dissolution, diffusion and migration with the maximum yield of 26.39%, which was enhanced by 40.45% and 156.96% compared with individual ultrasonic extraction (UE) and enzymatic extraction (EE). According to the modified Fick’s second law of diffusion, the extraction process of TFPs illustrated a good linear correlation (R^2^ ≥ 0.9), and the rate constant gradually elevated as the temperature increased from 25 to 45 °C, while the presence of ultrasound exerted a vital role in extracting TFPs. Regarding to the thermodynamic results, the positive values of ΔH and ΔG demonstrated that UAE, UE and EE were endothermic and unspontaneous processes. This study provides a theoretical basis for polysaccharide extraction processing.

## 1. Introduction

As a medicinal and edible fungus, *Tremella fuciformis* (*T. fuciformis*) is extensively cultivated in many countries, especially in China, Brazil and other East Asian countries [1,2]. *T. fuciformis* is becoming increasingly favored by consumers due to its tender texture and rich nutrients, primarily referring to polysaccharides, proteins, mineral elements and vitamins, endowing it with the characteristics of anti-antioxidant, anti-inflammatory, immune regulatory and hypoglycemic activities [3]. *T. fuciformis* polysaccharides (TFPs) are some of the most important substances that have garnered the most attention in recent years, and they have been widely explored in many aspects due to their high content (nearly 60% to 70% of the dry weight) and outstanding physiochemical and biological properties, such as their anti-freezing and self-healing ability [4], their ability to inhibit amylase digestion [5], antioxidant activity, stress resistance [6], etc.

Although TFPs are enriched with a series of promising functional properties in multiple areas, searching for an extraction method which is not only efficient and environmentally friendly but also could preserve the native quality as much as possible is still a matter of urgency. Until now, hot water extraction has been the most commonly used traditional method; it is generally conducted at a high temperature with a long duration, resulting in it being an energy- and time-consuming process with low yield [7]. Enzymatic-assisted extraction (EE) is a method that is mild and free from contamination, where the enzymes (mainly including cellulase, pectinase and proteases) can specifically disrupt the cell walls to promote the release of polysaccharides, and it can also maintain the quality of the extracts. Though it has good industrial application potential, the low output, long extraction time and high production cost are the shortcomings that need to be overcome [1]. It has been extensively proven that ultrasound as a green and non-thermal method is broadly applied in extraction processes, with the aim of elevating the extraction yield and reducing the time greatly by means of the collapse of cavitation bubbles to generate sufficient energy to bring about collisions between the particles [8]. Recently, multiple researchers have employed ultrasound in enzymatic extraction to provide an efficient way to produce an obvious improvement in the polysaccharide yield due to the thermal, mechanical and cavitation effects, promoting the probability of enzymes reacting with substrate to accelerate enzymatic hydrolysis as well as enzyme activity [9,10,11].

At present, the ultrasound-assisted enzymatic extraction (UAE) method has been widely used in the extraction of polysaccharides with multifarious merits, such as short extraction time, high yields of extracts and mild operating conditions. Furthermore, some studies have confirmed that low-frequency ultrasound is beneficial to the activation of cellulase within certain ultrasonic conditions [12,13,14,15]. Though reduced enzyme activity may have been found under some circumstances, the hydrolysis rate was mostly enhanced in the presence of ultrasound and cellulase [12]. Furthermore, in order to gain a better understanding of the complex diffusion, mass transfer and thermodynamic parameters that would influence the extraction process, more and more mathematical kinetic models are being constructed. From an engineering perspective, the establishment of these dynamic models helps to extract process design, optimization and control and provides useful information for expanding the industrial production scale [16]. Until now, few researchers have reported extraction kinetic and thermodynamic studies of TFPs.

Therefore, the combination of cellulase and ultrasound-assisted extraction would presumably be a more effective way to obtain TFPs. The objective of this study was to evaluate the effect of ultrasound and cellulase in extracting TFPs, and optimization was conducted. Furthermore, the extraction kinetics were established based on the modified Fick’s second law, and the thermodynamic kinetics were also investigated in order to provide a theoretical basis for the TFP extraction process in both experimental designs and in industrial applications. 

## 2. Materials and Methods

### 2.1. Materials 

*T. fuciformis* was purchased from a local supermarket in Jinan, Shandong Province, China. It was dried at 50 °C and ground to a fine powder with the particle size of 128 μm. Cellulase was purchased from Shanghai Macklin Biochemical Technology Co., Ltd. (Shanghai, China). All other reagents were of analytical grade.

### 2.2. Different Extraction Methods 

#### 2.2.1. Enzymatic Extraction (EE)

*T. fuciformis* powder and deionized water was mixed in a beaker with different solid–liquid ratios (1:60–1:110) and placed into a shaking water bath (SHZ-A, Shanghai Boxun Medical Biological Instrument Co., Ltd., Shanghai, China) at a preheated temperature (30–70 °C). Then, a certain amount of cellulase (1–5%, *w*/*v*) was added into the mixture and kept at a constant temperature for 60 to 210 min with a stirring rate of 150 r/min. After extraction, the mixture was immediately placed in boiling water for 10 min to inactivate the enzyme and then cooled and centrifuged at 4000 r/min for 15 min to obtain the supernatant.

#### 2.2.2. Ultrasound-Assisted Enzymatic Extraction (UAE) and Ultrasonic Extraction (UE)

*T. fuciformis* powder (3.0 g) was added into 270 mL of deionized water. It was extracted with cellulase (1–5%, *w*/*v*) and placed in an ultrasonic probe (SCIENTZ-IID, Ningbo Scientz Biotechnology Co., Ltd., Zhejiang, China) with the frequency of 20 kHz and a total power of 1200 W. The extraction temperature was controlled by a low-temperature thermostatic water bath (DC-1015, Ningbo Scientz Biotechnology Co., Ltd., Zhejiang, China). The ultrasonic treatment conditions were set as follows: ultrasonic power from 300 to 700 W, durations from 10 to 50 min, and temperature from 30 to 50 °C. After the steps of enzyme inactivation, cooling, and centrifugation, the supernatant was collected.

UE was selected as the reference process in the kinetics analysis, and the conditions were the same as those in the UAE without cellulase.

### 2.3. Determination of the Yield of TFP

The collected supernatant obtained by different methods was concentrated to one-third of its original volume according to a rotary evaporator at 60 °C under vacuum. The concentrate was precipitated with 95% (*v*/*v*) ethanol (four times the volume of the concentrate) and incubated for 4 h at 4 °C in refrigerator. The precipitant was centrifuged at 4000 r/min for 15 min and collected and freeze-dried to obtain TFP (the yield could reach as high as 34.53% under the hot water extraction process with the temperature of 90 °C for 14 h). The extraction yield of TFP was calculated as follows using Equation (1):(1)$$\mathrm{TFP}\;\mathrm{yield}\;(\%)=\frac{\mathrm{Weight}\;\mathrm{of}\;\mathrm{TFP}\;(g)}{\mathrm{Weight}\;\mathrm{of}\;\mathrm{Tremella}\;\mathrm{fuciformis}\;\mathrm{powder}\left(g\right)}$$

### 2.4. Response Surface Methodology (RSM) with Box–Behnken Design (BBD)

BBD with three levels and three factors was carried out to further optimize the UAE conditions, where the influencing factors included ultrasonic power (A: 500–700 W), duration (B: 30–50 min) and temperature (C: 40–50 °C). In order to achieve the best combination of the three variables, a total of 17 experimental points were conducted, including 5 center tests and 12 factorial tests (Table 1). The data were analyzed by Design Expert software (version 10.0).

### 2.5. Determination of Extraction Kinetic Parameters

The extraction of TFP is an unsteady-state diffusion process, no matter whether water or ultrasound is used, due to the solvent infiltration and internal and external diffusion, where the TFP concentration within the particles continues to decrease with the extension of the extraction time; therefore, it is quite suitable describe it by using Fick’s second law. Before applying this model, the following assumptions need to be made to simplify the complexity of the extraction process dynamic model appropriately:(a)The crushed *T. fuciformis* are uniform spherical particles, and the particle shape remains almost unchanged throughout the whole extraction process.(b)At the beginning of extraction and any sampling interval, the *T. fuciformis* particles are evenly distributed, and the mass transfer resistance of the particle surface can be ignored.(c)TFPs are evenly distributed in the *T. fuciformis* particles with radial diffusion (ignoring axial diffusion), and the mass concentration and diffusion coefficients of TFPs remain constant.(d)There are no chemical reactions or degradation of polysaccharides during the extraction process.
(2)$$\frac{\partial C}{\partial t}=D_{s}(\frac{\partial^{2}C}{\partial^{2}x}+\frac{\partial^{2}C}{\partial^{2}y}+\frac{\partial^{2}C}{\partial^{2}z})$$

The particles of *T. fuciformis* were regarded as spherical; Equation (2) could be converted to Equation (3).
(3)$$\frac{\partial C}{\partial t}=D_{s}(\frac{\partial^{2}C}{\partial r^{2}}+\frac{2}{r}\times \frac{\partial C}{\partial r})$$

The initial and boundary conditions were set as follows: r = 0, C = 0 and r = R.
(4)$$(\frac{\partial C}{\partial t})V={-D}_{s}\times A\times {\left(\frac{\partial C}{\partial r}\right)}_{r=R}$$
where R is the radius of TFP particles (mm), V is the volume of the solution (mL), D_s_ is the internal effective diffusion coefficient of TFPs (mm^2^/min) and A is the contact area between *T. fuciformis* particles and the extraction solvent (mm^2^). Equation (5) can be obtained by Fourier transformation from Equation (4) [17]:(5)$$\frac{C_{\infty }-C}{C_{\infty }-C_{0}}=\frac{6}{\pi^{2}}\sum_{n=1}^{\infty }\left\{exp\left[-{\left(\frac{n\pi }{R}\right)}^{2}D_{s}t\right]\right\}$$
where C_∞_ is the equilibrium mass concentration (mg/mL), C_0_ is the initial mass concentration (mg/mL) and t is the extraction time. For this system, n = 1 is usually a good approximation due to the distribution of the concentration being an infinite series, and its higher-order term tends to zero and can be ignored [18]. Equation (6) can be obtained as follows:(6)$$\frac{C_{\infty }-C}{C_{\infty }-C_{0}}=\frac{6}{\pi^{2}}exp(-\frac{\pi^{2}D_{s}t}{R^{2}})$$

When C_0_ is 0, Equation (7) can be obtained as follows:(7)$$\frac{C_{\infty }-C}{C_{\infty }}=\frac{6}{\pi^{2}}exp(-\frac{\pi^{2}D_{s}t}{R^{2}})$$

Equation (7) also can be converted to the logarithmic Equation (8) as follows:(8)$$\mathrm{Ln}(\frac{C_{\infty }}{C_{\infty }-C})=kt+ln[\frac{\pi^{2}\times C_{\infty }}{6\left(C_{\infty }-C\right)}]$$
where k is the extraction rate constant and k =$\;\frac{\pi^{2}\times D_{s}}{R^{2}}$.

Since $y=\frac{CV}{M}$, among them, y is the extraction yield of the TFPs, C is the concentration of polysaccharides in the extraction solution, M is the mass of raw material and V is the volume of the extraction solution. In this experiment, V and M are both constant values, so Equation (8) can be transformed into Equation (9) as follows:(9)$$\ln(\frac{y_{\infty }}{y_{\infty }-y})=kt+ln[\frac{\pi^{2}\times y_{\infty }}{6\left(y_{\infty }-y\right)}]$$

### 2.6. Determination of Thermodynamic Parameters

The relationship between the extraction rate constant and temperature of polysaccharides followed the Arrhenius equation, and the activation energy (E_a_) could be obtained according to the linear relationship between lnk and 1/T depicted by Equation (11) as follows:(10)$$k=Aexp(-\frac{E_{a}}{RT})$$
(11)$$lnk=lnA-\frac{E_{a}}{RT}$$

Furthermore, the thermodynamic parameters, including the changes in enthalpy free energy (∆H), entropy free energy (∆S) and Gibbs free energy (∆G), could be obtained by recording the relationship between ln(k/T) and 1/T according to Equations (12) and (13) as follows:(12)$$k=\frac{k_{B}T}{h}\exp\left(-\frac{∆G}{RT}\right)=\frac{k_{B}T}{h}\exp(-\frac{∆H}{RT}+\frac{∆S}{R})$$
(13)$$ln\frac{k}{T}=\frac{k_{B}T}{h}(-\frac{∆H}{RT}+\frac{∆S}{R})$$
where k_B_ and h are the Boltzmann constant (1.38 × 10^−23^ J/K) and Planck constant (6.6256 × 10^−34^ J/s), T is the extraction temperature (K) and k is the extraction rate constant (min^−1^).

### 2.7. Statistical Analysis

All the experiments were performed in triplicate, and data were represented as the mean value ± standard deviation. The figures were plotted using Origin Software Version 8.5 (Origin Lab Corp., Northampton, MA, USA), and statistical analysis was processed by ANOVA tests (*p* < 0.05) by SPSS 18.0 (SPSS Inc., Chicago, IL, USA).

## 3. Results and Discussion

### 3.1. Effects of Single-Factor Extraction Conditions on the TFP Yield

TFPs were extracted by cellulase under different conditions with and without ultrasound. Figure 1A displays the effect of the solid–liquid ratio on the yield of TFPs at a fixed temperature (50 °C), time (120 min) and enzyme concentration (3%); the TFP yield was enhanced as the ratio decreased from 1:70 to 1:90 and then became stable when it further elevated to 1:110. It was rational that the reduction in the ratio at the beginning could decrease the equilibrium concentration of TFPs, resulting in more polysaccharides being extracted into the solution. However, the further elevation in solvent led to a massive dilution in polysaccharide concentration and the contact between the polysaccharide and enzyme, thereby slowing down the increase in the extraction rate and resulting in significant losses during later treatment [19].

The number of enzymes exerted a great impact on the extraction yield. Figure 1B illustrates that the yield of TFPs increased as the enzyme concentration increased from 1% to 3% in both the EE and UAE process, which could have probably been due to the sufficient reaction between enzyme molecules and the substrates, thereby increasing the extraction yield [20]. In addition, the yield of UAE (27.81%) was two times higher than that of EE (11.04%), implying the cavitation effect of ultrasound severely damaged the sample tissue and cell wall, which favored the dissolution of intracellular polysaccharides. However, a too-high enzyme concentration, on the one hand, may not have been conducive to the movement of plant cells to the active site of the enzyme, and on the other hand, it was too excessive compared with the fixed liquid–material ratio, and it accordingly resulted in a reduction in enzymatic hydrolysis and lowered the extraction yield [21].

The effect of ultrasound power (0–700 W) on the yield of TFPs at 50 °C, 1:90 and 30 min is shown in Figure 1D. A significant rise in the TFP yield was found with an elevated power ranging from 0 to 600 W. The increase could be ascribed to the enhanced thermal, mechanical and cavitation effect as power boosted; it could not only damage the cell wall and the tissues of the raw material, facilitating the penetration of the solvents containing the enzymes, but could also accelerate the mass transfer rate and the binding of enzymes to substrates [22]. Nevertheless, no obvious further increase in the yield was obtained at the power of 700 W, which might have been attributed to the breakage of the molecular chains, the aggregation and viscosity alleviation of polysaccharides and the destruction of the enzyme structure at higher ultrasonic power, leading to the inactivation of the enzymes and rendering them inadaptable for the following extraction process [23]. 

Temperature plays a crucial role in extracting polysaccharides, especially in the presence of enzymes. The effect of different temperatures on the extraction yield in the EE (30–70 °C) and UAE (30–50 °C) process was investigated (Figure 1C). For the EE process, the TFP yield increased (from 9.02% to 10.26%) in the temperature range of 30–50 °C, whereas it declined when it was further enhanced to 70 °C. On the one hand, the optimal reaction temperature of cellulase was 50 °C, which was beneficial for maximizing the function of enzyme molecules to break the cell wall. On the other hand, a higher temperature could have resulted in a decrease in the solvent viscosity, facilitating the penetration of the solvent into the matrix particles, the release of the polysaccharides and the activation of enzymes, and consequently, it was beneficial to the yield of extraction. Nevertheless, excessive temperature resulted in the cushioning effect of cavitation, which would not only inactivate the enzymes but also depolymerize the polysaccharides, thereby damaging the TFP yield [24]. For the UAE process, the optimal extraction temperature was observed at 45 °C with a higher yield (27.71%); this temperature difference might be attributed to the thermal effect caused by ultrasound and the cushioning effect of cavitation at high temperature [25].

Extraction time is another important factor that would exert a great impact on the yield, and generally, the yield of polysaccharides is positively correlated with time within a certain time range [26]. As displayed in Figure 1E, the maximum yield of TFPs was obtained at 150 min (11.12%) and 50 min (26.73%) in the absence and presence of ultrasound, meaning that ultrasound could extremely shorten the extraction time and increase the extraction yield. As time prolonged, a decrease in the EE process was probable due to the degradation of the polysaccharides after an excessive extraction duration [23]. Due to the simultaneous generation of pressure shock waves and a high temperature coming from cavitation bubbles, the environment was no longer propitious to the active enzyme, and hence, the extension of ultrasonic time would not continually increase yields in the UAE process [27].

In brief, for the single-factor experiment, optimized experimental factors were selected: a liquid–solid ratio of 1:90, an enzyme concentration of 3% (*w*/*v*), ultrasound power of 500–700 W, a temperature of 40–50 °C and an extraction time of 30–50 min.

### 3.2. Model Fitting and Statistical Analysis by RSM

BBD with three factors and three levels was performed to optimize the mutual effect of three independent variables (ultrasonic power A, duration B and temperature C) on the extraction yield of TFPs. A total of 17 combinations were conducted to optimize the three independent parameters in the BBD (Table 1). The mathematical model was constructed by multiple regression analysis to explain the mutual interaction of the independent variables on the extraction yield of TFPs; the corresponding second-order polynomial equation was expressed as follows:$$Y=25.5+1.15A+2.01B+0.38C-0.17\mathrm{AB}-0.58\mathrm{AC}+0.20\mathrm{BC}+0.076A^{2}+0.11B^{2}-0.51C^{2}$$
where Y is the TFP extraction yield (%), and A, B and C are ultrasonic power, duration and temperature, respectively. 

The statistical ANOVA results are listed in Table 2. As two important factors, it is well known that the higher the *F*-value and the lower the *p*-value, the more significant the regression model will be. As shown in Table 2, the value of *F* (139.25) and *p* (<0.0001) manifested the significance of the regression model. In addition, the *p*-value (0.1365) and *F*-value (3.35) of the lack-of-fit demonstrated the high accuracy of the model [28]. The determination coefficient (R^2^ = 0.9944) implied that only 0.56% of the total variation was not explained by the model, and a closer adjusted determination (R^2^_Adj_ = 0.9873) not only further confirmed the significance of the model but also revealed that the obtained values were highly correlated with the predicted data from the regression model. Meanwhile, the excellent precision and reliability of the regression model could be displayed by the low value of the coefficient of variation (C.V. = 0.76%). Moreover, the linear coefficients (A, B and C), cross product coefficients (AC) and quadratic term coefficients (C^2^) were significant (*p* < 0.05), while others were insignificant (*p* > 0.05), and the dominating order of the influencing factors was duration > ultrasonic power > temperature. Hence, the model was accurate and reproduceable enough to predict the TFP extraction yield within the applied conditions in the period of the UAE process.

### 3.3. Effect of Parameter Interactions on TFP Extraction Yield

Figure 2 depicts the mutual effect of three variables on the TFP yield by 3D response surfaces (interpreting the mutual influences on the extraction yield of TFPs) and its corresponding 2D contour plots (manifesting the reciprocal interactions between test variables). Figure 2A illustrates the interactive effects of ultrasonic power and duration on the TFP yield at a fixed temperature of 45 °C. As the ultrasonic power or time elevated, the TFP yield enhanced firstly and then went flat, conveying a significant effect on the TFP extraction yield. The increase in yield was ascribed to the promotion of the cavitation effect on the dissolution of polysaccharides, that is, the enhancement in solubility and diffusion from plant tissue to extraction solvent as well as the enzyme activity, while the decrease might have been due to the degradation effect of excessive conditions on the extracted products. Compared to ultrasonic duration, power imposed a more significant quadratic effect on yield. Figure 2C demonstrates the quadratic effects of power and temperature on the yield of TFPs when the extraction time was fixed at zero. Similarly, the yield of TFPs was boosted with the increase in power, attributed to the accelerated penetration of solvents into the matrix, thereby promoting release, whereas as the power of ultrasound further enhanced, the raised cavitation effect imposed a side effect both on the enzymes and the mass transfer. Figure 2E shows that when ultrasonic power was fixed at zero, no significant quadratic effects of extraction duration and temperature on the yield were observed. It is noteworthy that the 2D contour plots shown in Figure 2B,D,F indicate that the interaction between temperature/power and duration was not significant (*p* > 0.05), which might mainly have been due to the selected parameter range.

### 3.4. Verification of Predictive Model

The applicability of the model equation to predict the optimal response value was examined by selecting the optimal conditions. According to Figure 2, the optimal conditions of UAE on TFP yield were as follows: ultrasonic power (A) of 698.159 W, duration (B) of 49.91 min and temperature (C) of 44.27 °C. A maximum response value of 28.62% was predicted. For the convenience of the operation and to confirm the suitability, the conditions were adjusted with slight modifications: ultrasonic power of 700 W, time of 50 min and temperature of 45 °C. The corresponding yield of 28.19 ± 0.42% was observed after triplicate experiments, which was nearly that of the predicted yield. Consequently, the model was valid enough to describe the UAE process.

### 3.5. Kinetics Analysis 

The kinetic parameters of the kinetic model, including the extraction rate constant (k_1_), relative raffinate rate (k_2_), surface diffusion coefficient (D_s_) and half-life period (t_1/2_), were obtained.

#### 3.5.1. Analysis of k_1_


The value of k_1_ indicates the dissolution rate of TFPs, and the higher the value, the faster the polysaccharide dissolution rate [29]. According to Equation (9), in order to obtain the kinetic parameters involved in the extraction kinetic model, the relationship of ln[(y_∞_)/(y_∞_ − y)] versus extraction time (t) is illustrated in Figure 3 and the linear regressions results are listed in Table 3. It was found that good linear fits of the UAE, UE and EE processes were observed (R^2^ > 0.9), implying that Fick’s second law could be appropriately applied in the extraction of TFP. In addition, the k_1_ values of different extraction process were all increased as the temperature enhanced from 30 to 45 °C, indicating that increased temperature was beneficial for the thermal movement of molecules and the dissolution of TFPs. Notably, the k_1_ of UAE was higher than that of the UE and especially the EE process, signifying that ultrasound, on the one hand, facilitated the swelling rates of *T. fuciformis* particles, and on the other hand, boosted the process of solvent penetration and internal and external diffusion [30].

#### 3.5.2. Analysis of k_2_

k_2_ displays the ratio of the undissolved polysaccharides to the concentration of the polysaccharides in the sample when the dissolution reaches equilibrium [29]. It could be obtained according to the exponential fitting of [(y_∞_)/(y_∞_ − y)] and time (t), and the relationship between k_2_ and extraction temperatures is shown in Figure 4. As illustrated in Table 4, _2_ was linearly related to temperature (R^2^ > 0.9) under all three extraction methods, further manifesting that the extraction process also fitted well with the exponential model. Similar to k_1_, k_2_ elevated as the temperature increased from 30 to 45 °C, and an alleviating trend occurred when it further increased to 50 °C. The difference between k_1_ and k_2_ could be attributed to the various fitting methods [31]. In conclusion, the established extraction kinetic model could exactly describe the TFP extraction process with UAE in this study.

#### 3.5.3. Analysis of D_s_

The equilibrium concentration indicates the limit of TFP diffusion and migration, and the migration speed is generally reflected by D_s_. D_s_ had a positive relation with k_1_ and the particle radius R, which could be obtained by the equation k_1_ = π^2^D_s_/R^2^. As shown in Figure 5A, a higher temperature caused faster diffusion and migration within a certain range (30–45 °C). Furthermore, the value of D_s_ of UAE and UE was higher than that of EE. This could be explained by the fact that the ultrasonic cavitation effect not only reduced the material particles but also promoted mass transfer [32]. The above results illustrated that ultrasound could accelerate the D_s_ of TFPs and shorten the extraction time [33].

#### 3.5.4. Analysis of t_1/2_

The value of half-life (t_1/2_ = ln2/k_1_) regarding the extraction efficiency normally refers to the time it takes for the concentration of a sample to decrease to one-half of its original value. The value of t_1/2_ illustrated a gradually reduced tendency as temperature increased from 30 to 45 °C, implying the increased temperature benefited the dissolution of TFPs and the enhancement of extraction efficiency. In addition, the values of UAE and UE, especially UAE, were significantly lower than that of EE, suggesting the extraction time could be shortened due to the reinforced fragmentation of the *T. fuciformis* cell wall generated by the ultrasonic cavitation effect, resulting in more TFPs to be dissolved in the solution [29,32].

### 3.6. Thermodynamic Parameters Analysis

The thermodynamic parameters could be obtained based on the rate constants, as shown in Table 5. E_a_ indicates the energy needed when molecules change from a normal state to an activated state, and in this study, it was applied to investigate the feasibility of the different extraction methods; commonly, the higher the extraction rate coefficient, the lower the energy required [34].

E_a_ was obtained according to lnk against 1/T, and the lower the value, the faster the reaction [35]. As shown in Table 5, the E_a_ of UAE was reduced by 26.42% and 65.48% when compared with the UE and EE process, suggesting that ultrasound played a main role in the extraction of TFPs and the EE process was more sensitive to the temperature than UE and UAE [23]. On the one hand, the mechanical and free radical effect was beneficial to the destruction of the cell wall, resulting in faster dissolution, and on the other hand, it broke through the energy barrier by accelerating the collision rate between enzyme molecules and substrates, therefore facilitating the cellulase hydrolysis reaction [9].

The other three thermodynamics parameters, including ΔH, ΔS and ΔG, could be determined based on Equation (12). The positive value of ΔH signified the endothermic nature of the extraction process [36]. In addition, ΔH declined by 63.20% and even 73.46% when ultrasound and ultrasound-assisted enzymes were applied in this extraction, implying that ultrasound activated plenty of particles from the original state, thereby alternating their conformation and reducing the energy requirement [37]. Moreover, more energy could be provided by the absorption and conversion of plenty of ultrasound energy [38]. In addition, a decreased ΔS was found when ultrasound existed in the extraction process, manifesting a raised ordered arrangement of the substrates and enzymes in the extraction system [35]. Furthermore, all of the three extraction processes could not occur spontaneously on the basis of the positive value of ΔG. Additionally, the reduced ΔG of UAE and UE compared with EE indicated the employment of ultrasound was beneficial to spontaneous processes; this result was consistent with the study of Zhao et al. [35]. Therefore, it could be concluded that the presence of ultrasound was beneficial to the extraction process.

## 4. Conclusions

UAE was applied in extracting TFPs, resulting in a higher extraction efficiency with less time consumption. The comparison of UAE, UE and EE extraction kinetics and the thermodynamics parameters based on Fick’s second law was studied, confirming the importance of ultrasound in TFP extraction. ANOVA analysis illustrated that in the selected range, ultrasonic power and time exerted significant influences on the extraction yield, and the optimal yield of 28.19 ± 0.42% was achieved by UAE with the extraction conditions of 700 W, 50 min and 45 °C. The results of the kinetic model, including UAE, UE and EE, revealed that ultrasound could not only facilitate the swelling rates of *T. fuciformis* particles but also boosted the process of solvent penetration and internal and external diffusion, resulting in a high yield of UAE (26.39%) and UE (18.79%) compared with EE (10.27%). The thermodynamic parameters of positive ΔH and ΔG demonstrated the endothermic and unspontaneous process of these three extraction methods. In conclusion, UAE was considered as the most efficient way to extract TFPs, providing meaningful guidance for the progress of polysaccharide extraction. Further studies regarding the mechanism of UAE in extracting TFPs are still needed to determine the physico-chemical and structural properties of the extracted TFPs.

## Figures and Tables

**Figure 1 foods-13-01408-f001:**
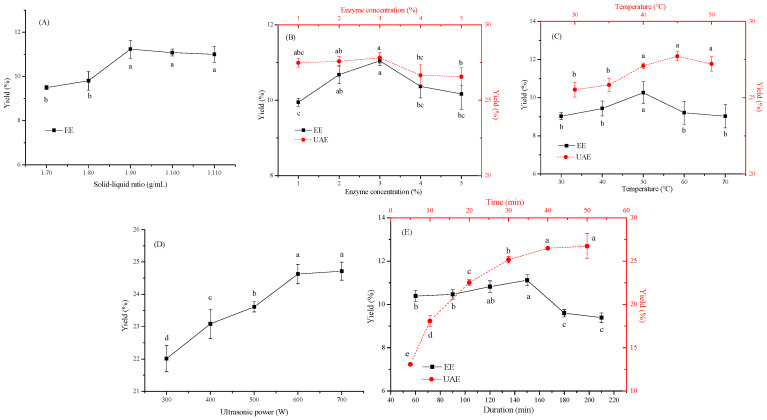
Effect of different extraction factors on extraction yield of TFP: (**A**) solid–liquid ratio, (**B**) enzyme concentration, (**C**) temperature, (**D**) ultrasonic power, (**E**) duration. Note: different lowercase letters (a–e) in the same figure indicate significant differences (*p* < 0.05).

**Figure 2 foods-13-01408-f002:**
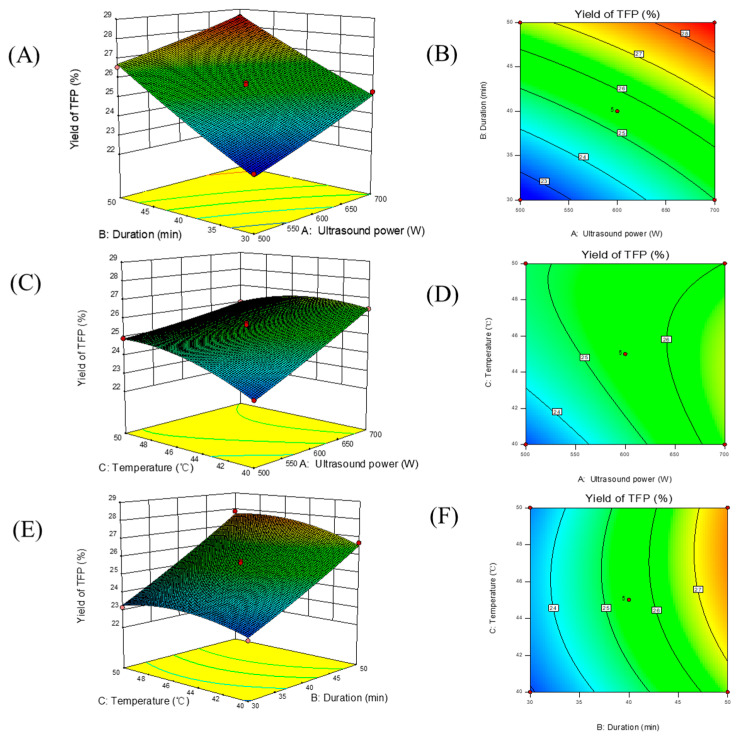
Three-dimensional response surfaces and the corresponding two-dimensional contour plots of the effects of ultrasound power, ultrasound time and temperature on the yield of TFP: (**A**,**B**) ultrasonic power and duration, (**C**,**D**) ultrasonic power and temperature, (**E**,**F**) duration and temperature.

**Figure 3 foods-13-01408-f003:**
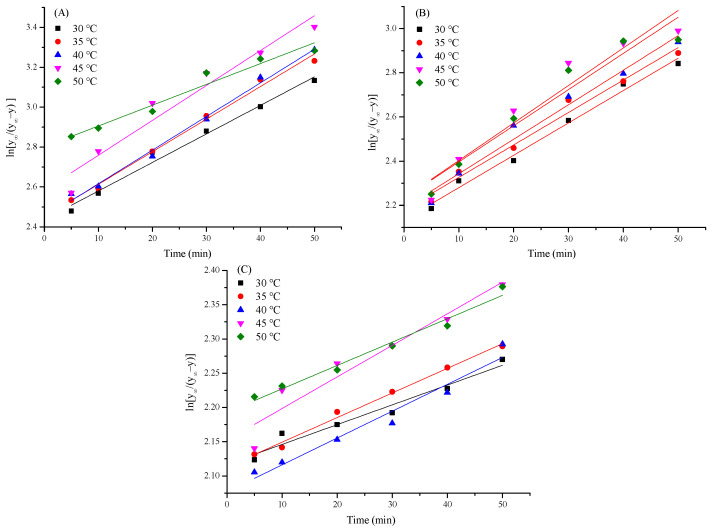
The relationship between ln[(y_∞_)/(y_∞_ − y)] and t with different extraction methods ((**A**): UAE, (**B**): UE, (**C**): EE).

**Figure 4 foods-13-01408-f004:**
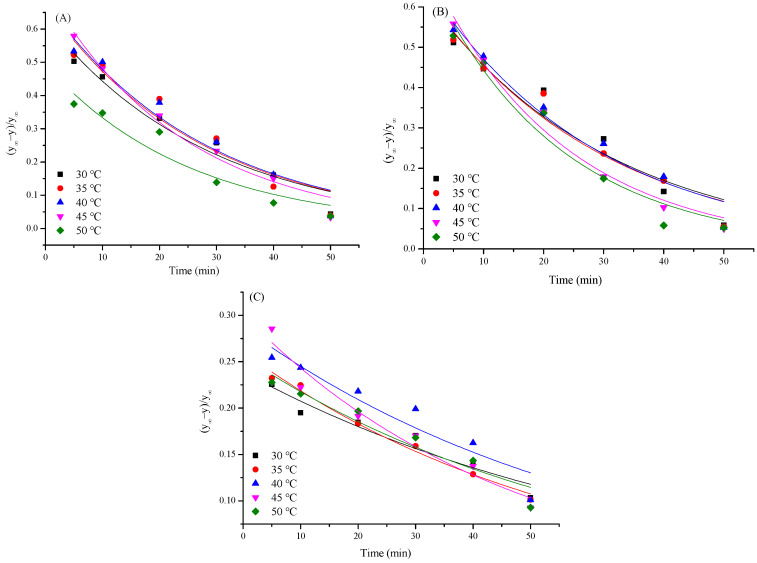
The relationship between (y_∞_ − y)/(y_∞_) and t under different temperatures ((**A**): UAE, (**B**): UE, (**C**): EE).

**Figure 5 foods-13-01408-f005:**
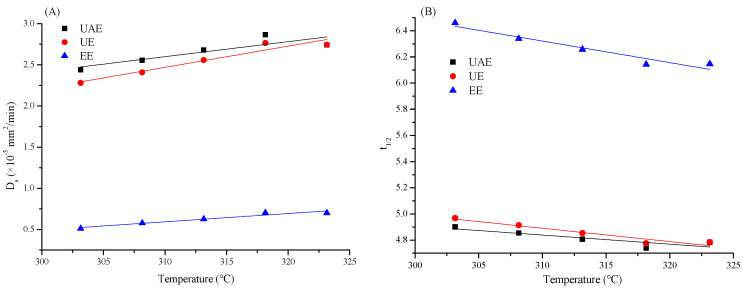
The relationship the relationship between the D_s_ and temperature (**A**) and between t_1/2_ and temperature (**B**) in three extraction process of TFP.

**Table 1 foods-13-01408-t001:** BBD with independent variables and experimental results.

Run Numbers	A: Ultrasonic Power (W)	B: Duration (min)	C: Temperature (°C)	TFP Yield (%)
1	700 (1)	40 (0)	50 (1)	25.99
2	600 (0)	40 (0)	45 (0)	25.59
3	600 (0)	30 (−1)	45 (0)	25.47
4	500 (−1)	50 (1)	45 (0)	26.54
5	500 (−1)	40 (0)	50 (1)	24.9
6	500 (−1)	40 (0)	40 (−1)	22.99
7	600 (0)	30 (−1)	50 (1)	23.13
8	500 (−1)	30 (−1)	45 (0)	22.5
9	700 (1)	50 (1)	45 (0)	28.54
10	600 (0)	40 (0)	45 (0)	25.39
11	600 (0)	50 (1)	50 (1)	27.85
12	600 (0)	40 (0)	45 (0)	25.37
13	700 (1)	30 (−1)	45 (0)	25.16
14	600 (0)	30 (−1)	40 (−1)	22.76
15	600 (0)	50 (1)	40 (−1)	26.67
16	700 (1)	40 (0)	40 (−1)	26.41
17	600 (0)	40 (0)	45 (0)	25.69

**Table 2 foods-13-01408-t002:** Variance analysis of parameters in regression equation.

Variables	Sum of Squares	df	Mean Square	*F*-Value	*p*-Value
Model	46.61	9	5.18	139.25	<0.0001
A	10.51	1	10.51	282.61	<0.0001
B	32.20	1	32.20	865.75	<0.0001
C	1.15	1	1.15	31.06	0.0008
AB	0.11	1	0.11	2.93	0.1308
AC	1.36	1	1.36	36.49	0.0005
BC	0.16	1	0.16	4.41	0.0739
A^2^	0.025	1	0.025	0.66	0.4425
B^2^	0.048	1	0.048	1.28	0.2945
C^2^	1.08	1	1.08	28.98	0.0010
Residual	0.26	7	0.037		
Lack of fit	0.19	3	0.062	3.35	0.1365
Pure error	0.074	4	0.019		
Cor. Total	46.87	16			

R^2^ = 0.9944, R^2^_Adj_ = 0.9873, R^2^_Pred_ = 0.9339, C.V. = 0.76%.

**Table 3 foods-13-01408-t003:** Linear regression of the relative extraction rate of TFPs extracted by different methods at different temperatures.

Extraction Methods	Temperature (°C)	Linear Equation	R^2^	y_∞_	k_1_ (min^−1^)	D_a_ (×10^−4^ mm^2^/min)
UAE	30	y = 0.0143x + 2.4369	0.9830	23.99	0.0143	2.44
	35	y = 0.0163x + 2.4501	0.9905	26.37	0.0163	2.55
	40	y = 0.0169x + 2.4478	0.9907	27.84	0.0169	2.68
	45	y = 0.0175x + 2.5841	0.9360	31.06	0.0175	2.86
	50	y = 0.0104x + 2.8015	0.9525	27.72	0.0104	2.74
UE	30	y = 0.146x + 2.1338	0.9866	18.21	0.0146	2.28
	35	y = 0.0147x + 2.1796	0.9773	19.04	0.0147	2.41
	40	y = 0.0156x + 2.1873	0.9698	19.96	0.0156	2.56
	45	y = 0.0169x + 2.2332	0.9176	20.94	0.0169	2.77
	50	y = 0.0164x + 2.2328	0.9242	20.16	0.0164	2.74
EE	30	y = 0.0029x + 2.1172	0.9473	10.79	0.0029	0.51
	35	y = 0.0036x + 2.1134	0.9907	10.98	0.0036	0.58
	40	y = 0.0039x + 2.0769	0.9519	11.01	0.0039	0.63
	45	y = 0.0046x + 2.1523	0.9136	11.90	0.0046	0.70
	55	y = 0.0034x + 2.1930	0.9729	11.86	0.0034	0.70

**Table 4 foods-13-01408-t004:** Non-linear regression of the relative extraction rate of TFPs by different methods at different temperatures.

Extraction Methods	Temperature (°C)	Exponential Equation	R^2^	k_2_
UAE	30	y = 0.6263exp(−0.3461x)	0.9434	0.0346
	35	y = 0.6764exp(−0.0358x)	0.9095	0.0358
	40	y = 0.6824exp(−0.0356x)	0.9302	0.0356
	45	y = 0.7241exp(−0.0409x)	0.9719	0.0409
	50	y = 0.4933exp(−0.0392x)	0.9133	0.0392
UE	30	y = 0.6328exp(−0.0329x)	0.9122	0.0329
	35	y = 0.6378exp(−0.0338x)	0.9361	0.0338
	40	y = 0.6673exp(−0.0348x)	0.9584	0.0348
	45	y = 0.7025exp(−0.0447x)	0.9795	0.0447
	50	y = 0.7014exp(−0.0459x)	0.9545	0.0459
EE	30	y = 0.2390exp(−0.0141x)	0.9165	0.0141
	35	y = 0.2611exp(−0.0177x)	0.9863	0.0177
	40	y = 0.2871exp(−0.0158x)	0.9098	0.0158
	45	y = 0.3013exp(−0.0214x)	0.9435	0.0214
	50	y = 0.2557exp(−0.00161x)	0.9156	0.0161

**Table 5 foods-13-01408-t005:** The thermodynamics parameters in TFP extraction process by different methods.

Extraction Methods	E_a_ (kJ/mol)	∆H (kJ/mol)	∆S (J/mol)	∆G (kJ/mol) (318.15 K)
UAE	8.25	5.66	−261.19	88.76
UE	10.43	7.85	−254.74	88.89
EE	23.90	21.33	−223.14	92.31

## Data Availability

The original contributions presented in the study are included in the article, further inquiries can be directed to the corresponding author.

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
