# Peer review of "Study on the Optimization, Extraction Kinetics and Thermodynamics of the Ultrasound-Assisted Enzymatic Extraction of Tremella fuciformis Polysaccharides"

_foods, 2024, doi:10.3390/foods13091408_

Round 1

Reviewer 1 Report

Comments and Suggestions for Authors

The manuscript ID: foods-2962311 entitled “Study on the optimization, extraction kinetics and thermodynamic of ultrasound-assisted extraction of Tremella fuciformis polysaccharides by Hou et al.

  Specific comments      

The study was focused on the investigation of the effect of ultrasound and cellulase in extracting Tremella fuciformis polysaccharides (TFP) carried out at different extraction parameters. The influence of experimental conditions on the extraction yields of TEP was evaluated and optimized using response surface methodology (RSM).  The calculated optimal conditions for TFP extraction were found to be: an ultrasonic  power of 700 W, the temperature of 45 , and the time of 50 min. Furthermore, the extraction kinetics based on the modified Fick’s second law confirmed that the dissolution, diffusion and migration were significantly promoted by ultrasound-assisted enzymatic extraction (UAE). The UAE approach allowed significantly to enhance the maximum yield of TFP compared with ultrasonic extraction (UE) and enzymatic extraction (EE). Additionally, the extraction process of TFP was described by good linear correlation. Moreover, the thermodynamic parameters of positive ΔH and ΔG demonstrated an endothermic and unspontaneous process of these three extraction approaches. Thus, UAE can be considered as the most efficient extraction protocol for TFP, and the presented study providing an important guidance for the progress of polysaccharide isolation. The topic of the study can be interesting for the readers, the manuscript is well written. 

Author Response

On behalf of my co-authors, we thank you very much for giving us an opportunity to revise our manuscript. We appreciate editor and reviewers for their positive and constructive comments and suggestions on our manuscript entitled " Study on the optimization, extraction kinetics and thermodynamic of ultrasound-assisted enzymatic extraction of Tremella fuciformis polysaccharides " (Manuscript ID: foods-2962311). 

We would like to express our great appreciation to you and reviewers for comments to our paper. Looking forward to hearing from you!

Thank you and best regards.

Yours sincerely,

Furong Hou

Address: Institute of Agro-Food Sciences and Technology, Shandong Academy of Agricultural Sciences, 202 Gongye North Road, Jinan, Shandong 250100, China.

Reviewer 2 Report

Comments and Suggestions for Authors

The manuscript is interesting and well written.

Its main strength is the suucinct and extensive experimental pipeline tha has been implemented thus supporting the findings.

It addresses a simple issue efficiently but lacks a thorough discussion part.

The authors should add more comments in a separate discussion part.

Comments on the Quality of English Language

Minor editing of English language required

Author Response

On behalf of my co-authors, we thank you very much for giving us an opportunity to revise our manuscript. We appreciate editor and reviewers for their positive and constructive comments and suggestions on our manuscript entitled " Study on the optimization, extraction kinetics and thermodynamic of ultrasound-assisted enzymatic extraction of Tremella fuciformis polysaccharides " (Manuscript ID: foods-2962311). According to the comments raised by the reviewers, we gave the corresponding responses and made revisions using the “Track Changes” function in the manuscript.

We would like to express our great appreciation to you and reviewers for comments to our paper. Looking forward to hearing from you!

Thank you and best regards.

Yours sincerely,

Furong Hou

Address: Institute of Agro-Food Sciences and Technology, Shandong Academy of Agricultural Sciences, 202 Gongye North Road, Jinan, Shandong 250100, China.

Here are the responses of the comments, please see the attachment.

Reviewer 3 Report

Comments and Suggestions for Authors

This manuscript is a valuable contribution to the field, describing a complex research that provides a theoretical basis for the polysaccharide extraction process.The study extends the knowledge on the application of ultrasound and cellulase in polysaccharide extraction from Tremella fuciformis, as well as the optimization process to provide an improved solution for experimental design or industrial applications.

The introduction is consistent, the literature review has been critically followed in line with the state of the art.

The purpose of the research is clearly stated, the objectives are reasonable, well defined and achievable.

The study is well planned and the working methods are adequate to achieve the proposed objectives. The research methodology is well detailed.

The results are clearly presented and well-illustrated in a number of 5 Figures and 5 Tables. Please rename section 3. Results and discussions, instead of Results, as there is no separate discussion section.

The results obtained are extensively discussed and relevant explanations are provided, with the findings well highlighted. Some improvements need to be made in the discussions to better highlight the added value of the research.

The conclusions support the results of this study. This section needs to be improved, with a more concise/synthetic formulation, without repeating the results obtained, just showing how this research answers the questions asked and what are the prospects for further study.

The references are relevant to this research topic.

Some aspects can be improved:

- The genuine value and innovation of this study is not sufficiently highlighted.

- What are the limitations of this approach?

- Improving the conclusions.

The study has high application potential, as the combination of cellulase and ultrasound-assisted extraction has proven to be a more efficient way to obtain TFP.  Since it is a high-quality and high-impact approach to the field, I recommend its publication after a minor revision.

Author Response

(The authors gave the same response as above.)

Reviewer 4 Report

Comments and Suggestions for Authors

The main question addressed by the research is the exploration of the potential of ultrasound-assisted enzymatic extraction in intensifying the extraction yield of Tremella fuciformis polysaccharides. The study aims to investigate the effects of different extraction parameters on the extraction yields, optimize these conditions using RSM, and analyze the kinetic and thermodynamic aspects of the extraction process. The research also compares the extraction efficiency of this approach with individual ultrasonic extraction and enzymatic extraction methods.

This combination of techniques is not extensively explored in the literature, especially concerning Tremella fuciformis polysaccharides extraction. The present paper contributes to filling the gap in this field, but the major revision is required.

My suggestions are listed below:

- Please indicate in title "ultrasound-assisted enzymatic extraction" instead "ultrasound-assisted extraction"

- Line 88. Avoid the beginning of the sentence with the number. Rewrite to: T. fuciformis powder (3 g) was added into 270 mL deionized water.

- Which optimization approach you used for the optimization od enzymatic extraction, where you varied the solid-liquid ratio and enzyme concentration?

- Why you not include these two parameters in BBD?

- Figure 1 and figure 3 are not readable.

- You monitored the one response by this experimental design. My opinion is that you need to include at least two responses, with one or more functional properties... in order to correlate with the content of target bioactive compound.

- In this work, you applied the innovative extraction methods. It will be significant to perform some conventional extraction technique, commonly used for polysaccharide extraction (e.g., traditional hot water extraction) to provide a more comprehensive comparison of extraction efficiency.

Author Response

(The authors gave the same response as above.)
